# Medical student research productivity and scholarly impact: A 20-year bibliometric comparison with medical residents

Christian J. Hausner[1], Michelle Yi[1], Meet S. Patel[1], Jack T. Franchino[2], Charles S. Day[1,2,3]*

1 School of Medicine, Wayne State University, Detroit, Michigan, United States of America, 2 College of Human Medicine, Michigan State University, East Lansing, Michigan, United States of America, 3 Department of Orthopaedic Surgery, Henry Ford Health, Detroit, Michigan, United States of America

* cday9@hfhs.org

## Abstract

Research experience during medical school is widely recognized as a valuable component of medical education and an increasingly emphasized aspect of residency applications. However, growing research productivity among medical students has led to concerns regarding the quality and translatability of the research. To assess the impact of medical student-authored research, we compared it with resident-authored research across a 20-year period using a cohort study and bibliometric analysis of 1,443 student-authored and 5,365 resident-authored PubMed-indexed articles published from 2003 to 2023. NIH iCite Relative Citation Ratio (RCR), a field-and-time normalized measure of article influence, was utilized to quantify research quality. We identified publications authored by medical students or residents using standardized affiliation-based queries. Primary outcomes included publication volume, citation count, and RCR. Secondary analyses compared article impact by study design, specialty, and geographic origin. We found that medical student publications increased 15-fold from 2003 to 2023 (from <10 to ~150 per year). Moreover, despite publishing less overall volume than residents, medical student publications demonstrated greater scholarly impact than medical resident publications over this period (RCR 0.47 vs 0.26; p = 0.02). By 2023, the median relative citation ratio of medical student-authored publications was 0.8, approaching the benchmark for a median NIH-funded article (RCR = 1.0). Medical students published nearly twice as many articles in surgical specialties as medical specialties but with comparable median RCRs (0.44 vs 0.47; p = 0.73). Additionally, the median RCR of both U.S. and international medical student publications significantly increased over the past two decades. In conclusion, medical student research output has grown substantially, and our analysis demonstrates that medical students contribute impactful research that is frequently cited and built upon. These findings underscore the importance of student-led

**Data availability statement:** All relevant data are within the manuscript and its Supporting information files.

**Funding:** The author(s) received no specific funding for this work.

**Competing interests:** The authors have declared that no competing interests exist.

scholarship and continued support for medical student research opportunities. To facilitate future benchmarking of trainee research, we also provide *ImpactLens*: https://impactlensgit.netlify.app/.

## Introduction

Research experience during medical school is widely recognized as a valuable component of medical education, fostering a deeper understanding of evidence-based medicine and strengthening critical thinking skills [1]. Research has also become an increasingly emphasized metric for postgraduate training competitiveness [1]. A national survey of United States residency program directors (PD) in 2021 found that 41% of PDs said meaningful research participation will be more important in offering interviews considering the transition to pass/fail scoring for the United States Medical Licensing Examination (USMLE) Step 1 in 2022 [2]. Additionally, PDs from more competitive specialties were significantly more likely to value research than PDs from less competitive specialties [2]. This shift in what is valued on a residency application has led to a surge in research activities among medical students, with National Resident Matching Program (NRMP) data reporting a 138% increase in the mean number of research items (abstracts, presentations, and publications) among matched US MD seniors between 2014 (4.2) and 2024 (10.0) [3,4].

This upward trend in research engagement does not show signs of stopping, as 60% of US medical students now plan to redirect time previously spent preparing for USMLE Step 1 towards pursuing more research activity [5]. Likewise, medical residents face increasing demands to publish during residency [6]. The Accreditation Council for Graduate Medical Education (ACGME) 2025 guidelines mandate resident participation in scholarly activity, a requirement which, paired with intense clinical workload, places substantial strain on new residents to balance training, personal wellness, and career development [7–9]. Many studies have explored this growing body of medical trainee research [10–14], highlighting publication trends among medical students and residents, with less emphasis on the value or impact of their research. This may reflect the influence of prior studies such as a 2013 publication by Wickramasinghe et al., which found that from 1980–2010, 59% of medical student-authored articles were never cited [15]. Thus, there exists a paucity in recent literature examining whether such increased medical research output translates into meaningful scholarly impact.

Therefore, this study aims to address one fundamental question: has the quality of medical student research kept pace with the observed increase in research quantity? Using citation-based metrics, including the Relative Citation Ratio (RCR) [16], this study analyzes trends in trainee-authored publications from 2003–2023 by study design, medical specialty, and geographic region. Our primary objective was to compare the citation impact of medical student and resident research to assess whether student-authored publications demonstrate comparable scholarly influence. Secondary objectives included (1) examining temporal, specialty-specific, and international

trends in medical student research; and (2) developing specialty and country-specific RCR benchmarks to enable more objective evaluation of trainee research quality via *ImpactLens*. We hypothesized that medical student–authored research would demonstrate similar citation impact to resident-authored work and that standardized benchmarking would provide a reproducible framework for assessing the scholarly value of trainee publications across disciplines and regions.

## Materials and methods

A bibliometric analysis was conducted for medical student- and resident-authored research articles published between 2003 and 2023. Publications were identified through a PubMed search using affiliation-based queries for medical students and residents. This methodology was adapted from previously published bibliometric analyses of medical student research to enhance reproducibility and comparability (Wickramasinghe 2013, Elliot 2023) [9,15]. To create mutually exclusive article cohorts, PubMed Identifiers (PMID) from articles authored by both medical students and residents were arrayed and cross-referenced with Microsoft Excel. Two independent reviewers screened titles and affiliations to confirm author type (resident or medical student) and assess article eligibility. Articles without affiliation terms matching those in our query were excluded. This workflow is summarized in (S1 Fig). The original PubMed queries and lists of medical student and resident articles can be found in (S3 and S4 Appendices).

To evaluate research impact, PMIDs from each cohort's search results were uploaded into the National Institutes of Health (NIH) iCite tool (https://icite.od.nih.gov), a validated resource for citation analysis that provides volume and citation metrics including total publications, publications per year, total citations, median citations per publication, and the Relative Citation Ratio (RCR) – a field-and-time-normalized measure of article influence, with the median NIH-funded article scoring a 1.0 [16]. Of note, only articles from 2003–2023 could be included, since more recent publications are assigned a "projected" RCR derived from limited early citation data, which may not reflect their eventual impact. Mean, median, and weighted RCR were collected for both the resident and medical student cohorts to characterize the overall influence of medical student publications. Article metadata such as research focus, number of authors, presence of a medical student author, study type, study location, and institution, were manually obtained from PubMed. Article specialty was assigned based on the journal's primary specialty when it was singular, or if the journal covered multiple specialties, the specialty listed for the corresponding author was used. Article location was determined from the location stated by the authors, or, if not directly indicated, by the institutional affiliation of the corresponding author. The primary outcomes included the volume and scholarly impact of publications, as measured by citation counts and Relative Citation Ratio (RCR). Secondary outcomes included trends in study design, specialty, and geographic origin.

Data normality was assessed using the Shapiro-Wilk test. Given the non-normal distribution of citation and RCR data, non-parametric tests were employed for continuous variables. Mann-Whitney U tests were used to compare median values between groups (e.g., citation data for medical student vs. residents, surgical vs. medical specialties, U.S. vs. international publications). Fisher's exact tests were utilized to compare proportions between categorical variables (e.g., % of articles cited in a clinical document, % of articles with 0 citations, and % of articles in each impact factor tertile). Univariable linear regression was performed to assess trends in RCR over the 20-year period and correlation between specialty scholarly output and RCR across medical and surgical specialties, respectively. Specialty weighted scholarly impact was calculated to account for publication volume and RCR. This was done using the equation ($weighted\,impact = publication\,volume * median\,RCR$) where each specialty analyzed was plotted for visual comparison. Statistical significance was set at $p < 0.05$ for all analyses and Dunn's test was applied as a correction for multiple comparisons. There was no formal power analysis given the large sample size and exploratory nature of this study. Since this study did not utilize any human participants and all data was publicly available, IRB approval was not required. The data was analyzed using GraphPad Prism Version 10 for Windows (GraphPad Software, San Diego, California). This study was conducted and reported in accordance with the Strengthening the Reporting of Observational Studies in Epidemiology (STROBE) reporting guidelines for cohort studies.

## Results

PubMed searches yielded 1,815 publications authored by U.S. and international medical students, and 6,920 publications authored by medical residents from 2003 to 2023. After completing article screening, the final dataset included 1,443 unique articles authored by medical students and 5,365 unique articles authored by residents. For medical students, annual publication counts rose from fewer than 10 articles per year in 2003 to nearly 150 per year by 2023 – a 15-fold increase in volume (Fig 1a). These student-authored publications generated an average of 16.04 citations per article (SEM = 1.94), with a median of 4 citations and a maximum of 2,318 citations. The median article RCR was 0.47, indicating performance below the average NIH-funded publication in its field, and the yearly median RCR increased linearly over time (RCR = 0.0213 × Year, $R^2$ = 0.552; Fig 1b). By 2023, the median RCR for student-authored publications approached 1.0, suggesting comparability in impact with NIH-funded benchmarks and reflecting meaningful scholarly impact. The top ten most-impactful medical student-authored articles in our dataset can be found in (Table 1). These articles included

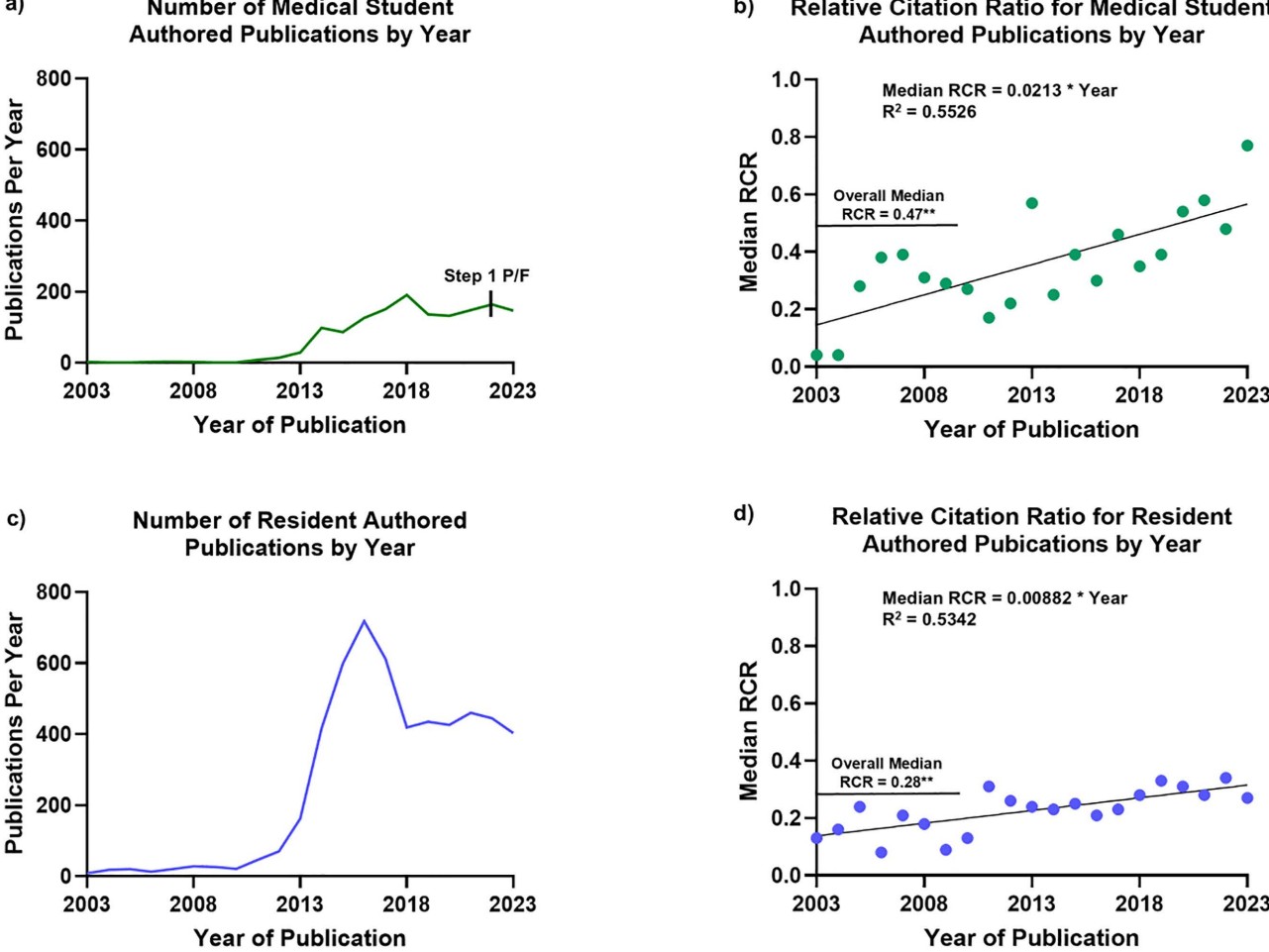

**Fig 1. Medical students and residents as knowledge contributors: Academic output and influence (2003-2023).** (a) Research output from medical student-authored publications (2003–2023). (b) Research impact in Relative Citation Ratio (RCR), a field-and-time normalized metric of article citations. (c,d) Corresponding data from resident-authored publications over the same period. Note the significant increase in publication volume and RCR for both groups. Despite contributing fewer publications overall, medical student-authored research generated a higher median RCR, with greater growth over time compared to resident-authored work.

**Table 1. Top 10 cited medical student-authored articles (2003-2023).**

| Article | Journal | RCR | Location | Specialty[a] | Study Type[b] |
|---|---|---|---|---|---|
| T cells expressing CD19 chimeric antigen receptors for acute lymphoblastic leukaemia in children and young adults | Lancet | 67.69 | United States | Oncology | Clinical Trial |
| Association Between Initial Use of e-Cigarettes and Subsequent Cigarette Smoking Among Adolescents and Young Adults | JAMA Pediatrics | 46.40 | United States | Pediatrics | Systematic Review |
| Preventing 30-day hospital readmissions | JAMA Internal Medicine | 26.28 | United States | Internal Medicine | Meta Analysis |
| Effect of Multimodal Prehabilitation vs Postoperative Rehabilitation on 30-Day Postoperative Complications for Frail Patients Undergoing Resection of Colorectal Cancer | JAMA Surgery | 25.49 | Canada | Colorectal Surgery | Clinical Trial |
| Suicide Rates After Discharge From Psychiatric Facilities | JAMA Psychiatry | 22.80 | United States | Psychiatry | Meta Analysis |
| Approximately One In Three US Adults Completes Any Type Of Advance Directive For End-Of-Life Care. | Health Affairs (Millwood) | 21.91 | United States | Internal Medicine | Meta Analysis |
| Harassment and discrimination in medical training | Academic Medicine | 21.49 | Canada | Obstetrics and Gynecology | Meta Analysis |
| What are the most common conditions in primary care? | Canadian Family Physician | 19.44 | Canada | Family Medicine | Systematic Review |
| Ipilimumab Therapy in Patients With Advanced Melanoma and Preexisting Autoimmune Disorders | JAMA Oncology | 16.44 | United States | Oncology | Multicenter Study |
| Quality of Life and Burnout Rates Across Surgical Specialties | JAMA Surgery | 14.82 | United States | Plastic Surgery | Systematic Review |

Relative Citation Ratio (RCR) reflects article-level influence normalized to NIH-funded publications, where a value of 1.0 represents the median influence of all NIH-funded papers in the same field.

[a]Specialty assignments were based on the primary clinical focus of the article and corresponding journal scope.

[b]Study type and location were determined from the publication metadata and abstract.

Table 1 top 10 most cited medical student-authored articles included in bibliometric analysis (2003–2023). Data are presented in descending RCR.

clinical trials, systematic reviews, and meta-analyses published in notable journals, including *The Lancet* and *JAMA*, and generated RCRs ranging from 14.82–67.69, all falling within the top 2% of NIH funded research. Notably, (635/1443) or 44% of indexed articles listed a medical student as the 1st author.

Medical residents published 5,365 unique articles during the same 20-year period, nearly four times the output of medical students (Fig 1c). These publications received an average of 8.38 citations per article (SEM = 0.33), with a median of 3 citations and a maximum of 591 citations. The median RCR was 0.28, significantly lower than that of medical student publications (p = 0.02, Fig 1b, d), although this is due, in part, to the types of articles being published (Table 2). While residents exhibited a more pronounced increase in publication volume, publishing over 50 times more articles in 2023 than in 2003, the growth in citation impact over time was more modest (RCR = 0.008 × Year, $R^2$ = 0.53; Fig 1d). Taken together, these findings suggest an improvement in medical trainee research volume and quality over the last two decades.

To contextualize these citation metrics, we next examined the study designs represented in medical student and resident authored publications. Among medical student publications, narrative reviews comprised the largest proportion of articles, accounting for 524 of 1443 publications (36.31%) (Fig 2a). Other common study designs included case reports (n = 442, 30.63%), systematic reviews (n = 134, 9.28%), observational studies (n = 133, 9.14%), randomized controlled trials (n = 81, 5.61%), meta-analyses (n = 65, 4.31%), multicenter studies (n = 40, 2.77%), and clinical trials (n = 17, 1.12%). Medical residents published a significantly higher percentage of case reports (n = 3435, 65% vs. n = 442, 31%) and a significantly lower percentage of narrative review (21% vs 36%) when compared to medical students (p < 0.001, Table 2). Citation performance varied by study design. Only meta-analyses achieved a median (RCR = 2.45) that significantly exceeded the

**Table 2. Comparison of study design between medical students and residents (2003-2023).**

| Study Design | Medical Students | Residents | p-value |
|---|---|---|---|
| Narrative Reviews | 524 (36%) | 1126 (21%) | **p < 0.001** |
| Case Reports | 442 (31%) | 3435 (65%) | **p < 0.001** |
| Systematic Review | 134 (9%) | 211 (6%) | p = 0.65 |
| Observational Study | 132 (9%) | 206 (3%) | p = 0.23 |
| Randomized Control Trial | 81 (6%) | 154 (3%) | p = 0.08 |
| Meta Analysis | 65 (4%) | 48 (1%) | **p = 0.03** |
| Multicenter Study | 40 (3%) | 101 (2%) | p = 0.42 |
| Clinical Trial | 17 (1%) | 84 (2%) | p = 0.41 |
| Total | 1443 (100%) | 5365 (100%) | – |

Table 2 comparison of study design between medical student- and resident-authored publications (2003–2023). Data are presented as counts and percentages. Narrative reviews and meta-analyses were proportionally more common among medical student publications, while case reports comprised most resident publications. Fisher's exact test was used to identify significant differences in the distribution of study types between the two cohorts.

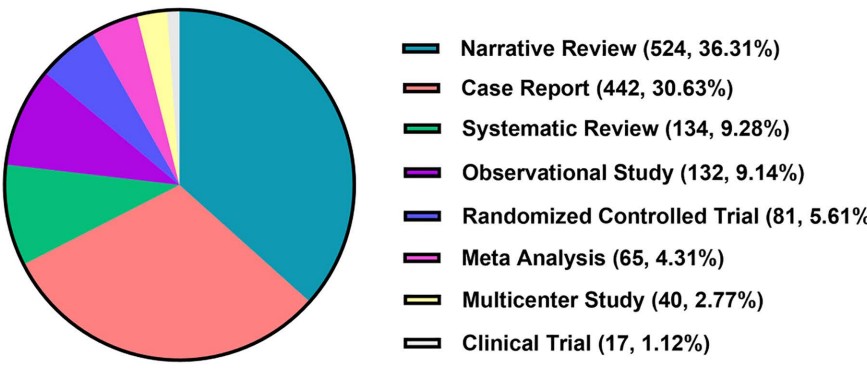

**a) Distribution of Medical Student Authored Research by Article Type**

- Narrative Review (524, 36.31%)
- Case Report (442, 30.63%)
- Systematic Review (134, 9.28%)
- Observational Study (132, 9.14%)
- Randomized Controlled Trial (81, 5.61%)
- Meta Analysis (65, 4.31%)
- Multicenter Study (40, 2.77%)
- Clinical Trial (17, 1.12%)

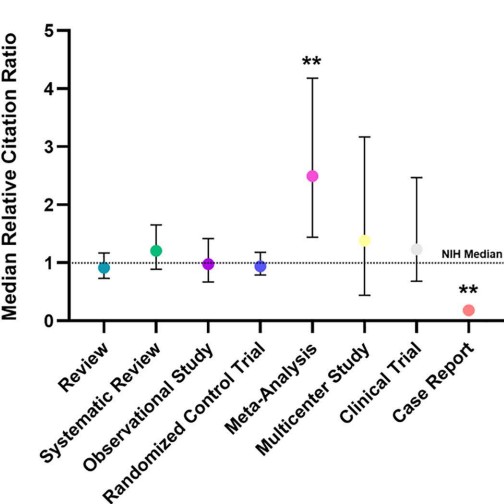

**b) Relative Citation Ratios by Article Type in Medical Student Authored Publications**

**Fig 2. Trends in study design among publications by U.S. medical students (2003-2023).** (a) Distribution of article types from U.S. medical student–authored publications (2003–2023), categorized into reviews, systematic reviews, observational studies, randomized controlled trials (RCTs), meta-analyses, multicenter studies, clinical trials, and case reports. Over half of all student-authored publications were reviews and case reports, highlighting a strong preference for lower-barrier entry points to scholarly output. (b) Median Relative Citation Ratio (RCR) by article type over the 20-year period. These findings emphasize that medical student research – regardless of study design – can achieve meaningful impact in the academic literature.

NIH benchmark of 1.0 (p < 0.001), while case reports were the only study design that fell significantly below an RCR of 1.0 (median RCR = 0.14, p < 0.001). Other study types, including systematic reviews, observational studies, and randomized control trials, had median RCRs near or slightly above 1.0 but did not differ significantly (Fig 2b). Exclusion of case reports in our analysis demonstrates that both medical student and resident research approach NIH benchmarks for scholarly impact, with median RCRs of 0.98 (n = 1001 articles) and 0.86 (n = 1,930 articles), respectively (Table 3).

Medical student-authored publications spanned both surgical (713 articles) and medical (730 articles) specialties (Fig 3). Articles in both surgical and medical fields generated a median of 4.00 citations per publication (p = 0.91, Fig 3a). The median

**Table 3. Comparison of research output between medical students and residents excluding case reports (2003-2023).**

| Metric | Medical Students | Residents | p-value |
|---|---|---|---|
| Total Pubs | 1001 | 1930 | – |
| Cites Per Pub (MED) | 11.00 | 9.00 | 0.57 |
| Relative Citation Ratio (MED) | 0.98 | 0.87 | 0.68 |

Table 3 descriptive volume and median citation metrics for medical student and resident-authored research demonstrates that both groups frequently publish non-case report articles, which approximate the median funded NIH publication (RCR = 1).

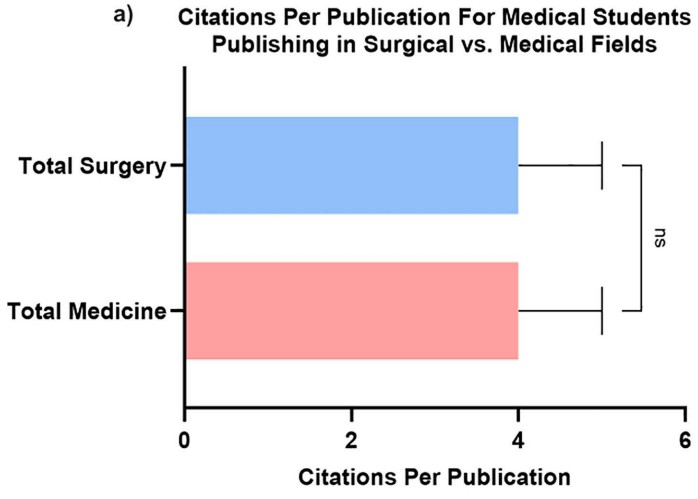

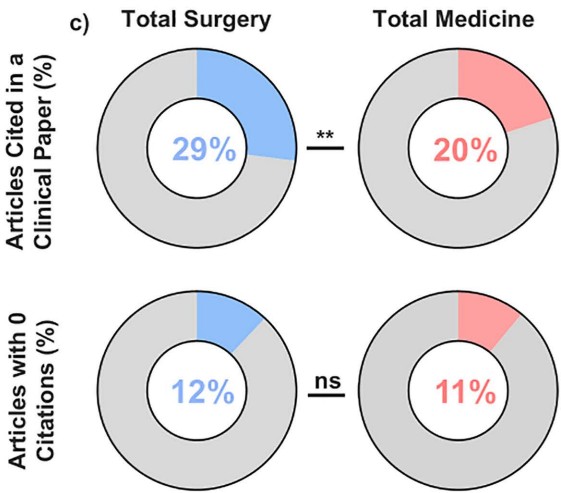

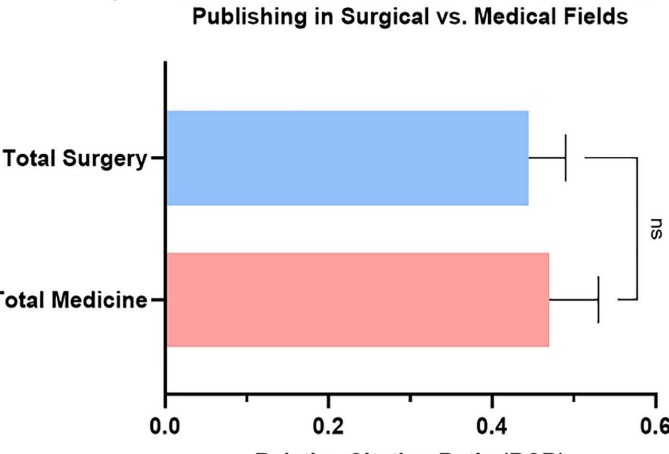

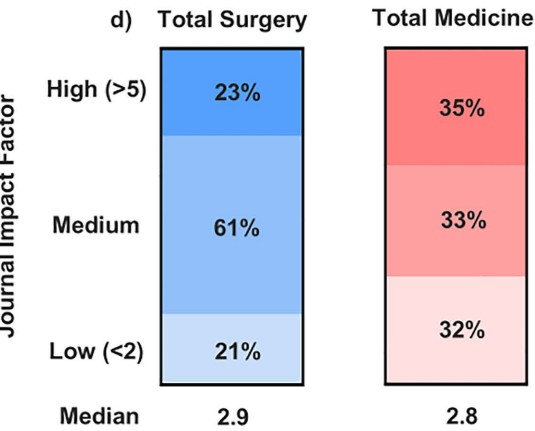

**Fig 3. Trends in medical student authorship by specialty with surgery vs. medicine (2003-2023).** (a,b) Compare group means in citations per publication and RCR. (c) Displays proportions of articles cited in clinical documents (top) and those with <10 citations (bottom). (d) Categorizes the impact factor (IF) of journals publishing medical student research into high (>5), medium, and low (<2) tiers.

RCR for surgical articles was comparable to medical articles (0.44 vs. 0.47, p=0.73, Fig 3b). A significantly greater proportion of surgical publications were cited in clinical documents (29% vs. 20%, p=0.02), yet the proportion of low-impact articles (with 0 citations) was similar between groups (surgery: 12%; medicine: 11%; p=0.83, Fig 3c) Medical articles were however significantly more likely to appear in high impact journals (35% vs. 23%, p=0.03), indicating a small but notable difference in visibility, although there was no overall difference in median impact factor (2.9 vs. 2.8, p=0.78, Fig 3d). The surgical specialties with the greatest number of medical student publications included: orthopedic surgery, obstetrics and gynecology, and neurosurgery. The medical specialties with the greatest number of medical student publications included: oncology, dermatology, and psychiatry (Fig 4). In benchmarking specialty-specific RCRs, the specialties with the highest median RCRs were pediatrics, oncology, and colorectal surgery (Fig 4). Simple linear regression of specialty output and median RCR yielded no significant correlation for either medical ($R^2$=0.07273) or surgical ($R^2$=0.08493) specialties (Fig 5). But overall, orthopaedic surgery and oncology were the specialties with the highest cumulative scholarly impact (Fig 6).

Comparison of U.S. and international medical student publications revealed distinct patterns in both output and impact. U.S. students published 697 articles from 2003 to 2023 with a median RCR of 0.58 (Fig 7a, b), while international students published 748 articles with a median RCR of 0.40 (p=0.02, Fig 7c, d). U.S. publications garnered more median citations per publication (5.00 vs. 3.00, p=0.01). Nonetheless, citation impact grew similarly between U.S. (RCR=0.01435×Year, $R^2$=0.71, Fig 7b) and international publications over time (RCR=0.01383×Year, $R^2$=0.66, Fig 7d), indicating that existing

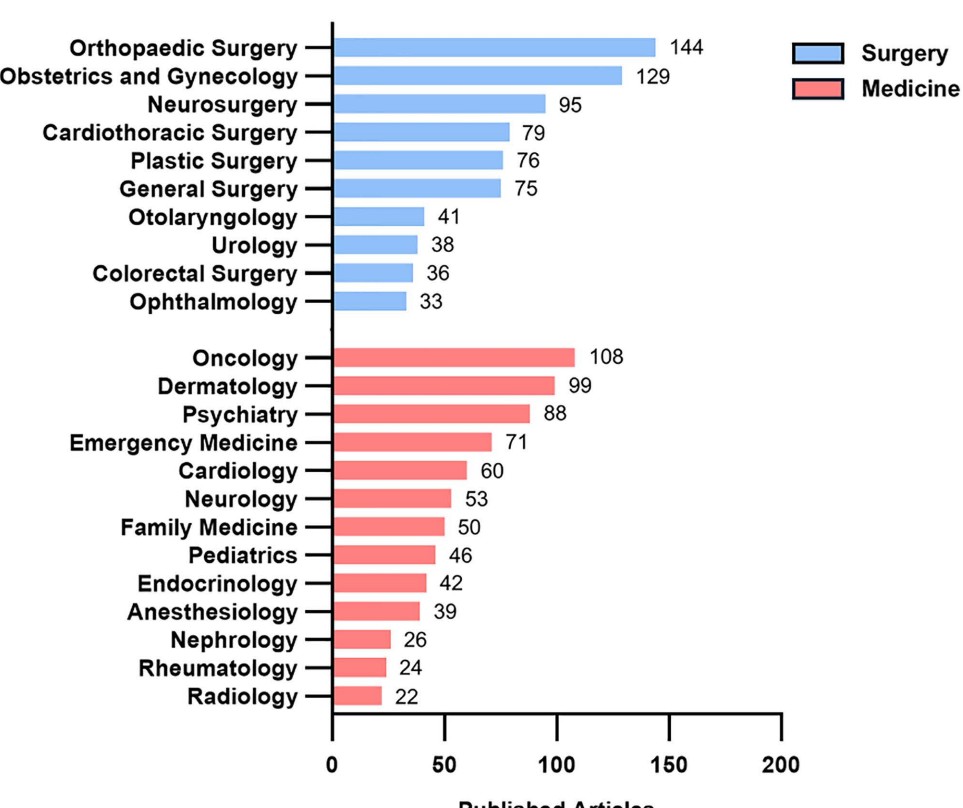

**Fig 4. Specialty publication trends in surgery vs. medicine among U.S. medical students (2003–2023).** Research output from U.S. medical student–authored publications (2003–2023), grouped by specialty and categorized into surgery-related and medicine-related fields.

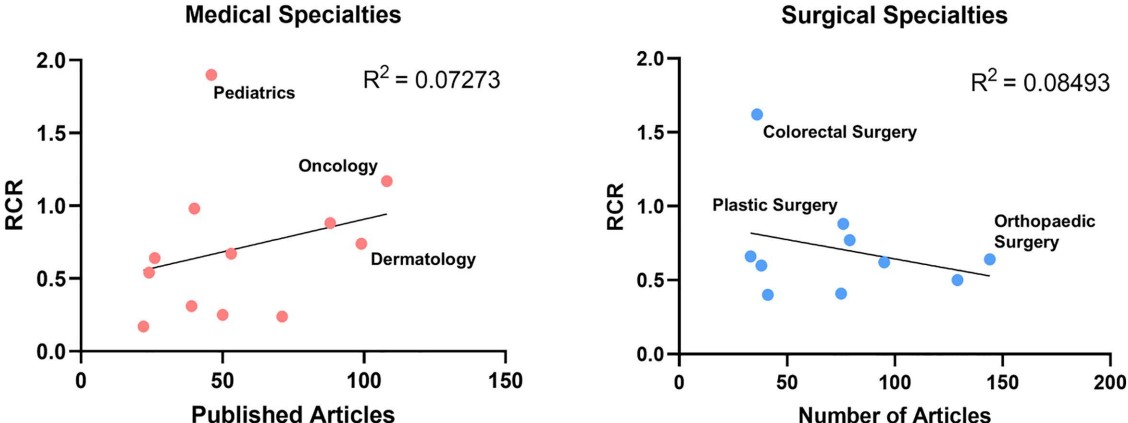

**Fig 5. There is relatively little correlation between specialty productivity and RCR.** Simple linear regression with specialty scholarly output and impact as measured by Relative Citation Ratio (RCR) shows no significant correlation for either medical subspecialties or surgical subspecialities.

### Citation Weighted Output Highlights Specialty-Level Differences in Scholarly Impact

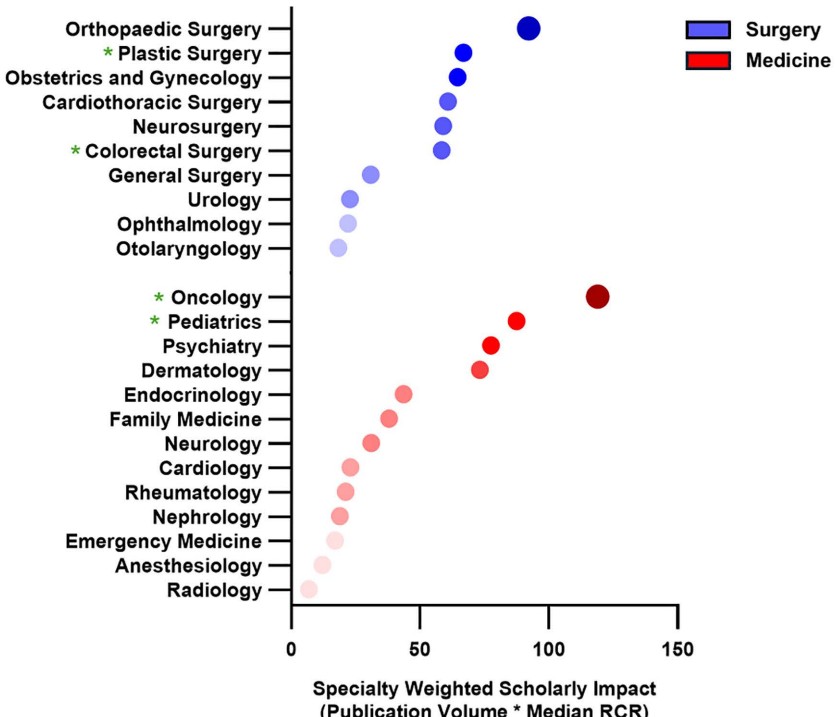

**Fig 6. Scholarly impact of medical student publications across specialties.** RCR-Weighted research output from U.S. medical student–authored publications (2003–2023), grouped by specialty and categorized into surgery-related and medicine-related fields. Specialties with the highest cumulative scholarly impact included orthopedic surgery and oncology. Plastic surgery, colorectal surgery, oncology, and pediatrics all outperformed their research output by volume alone with (median RCRs > 1).

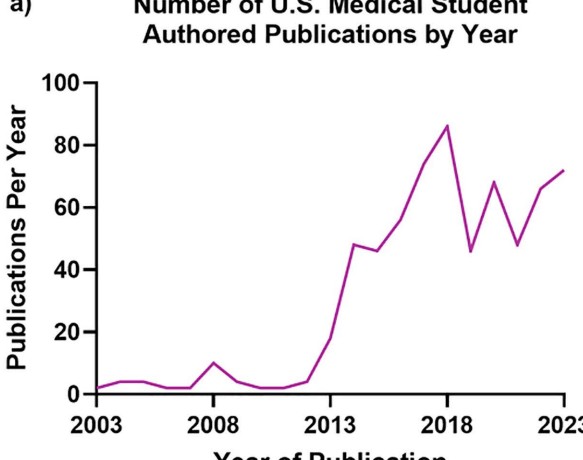

**a) Number of U.S. Medical Student Authored Publications by Year**

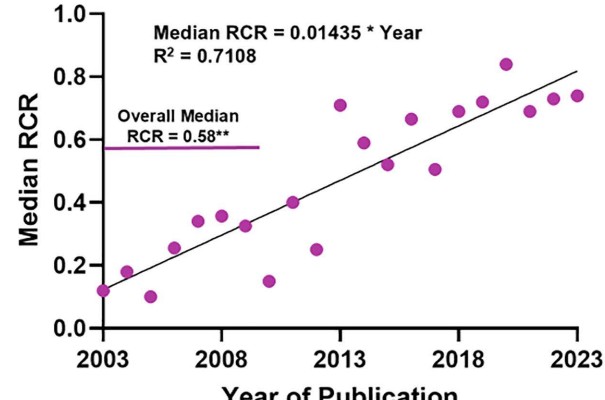

**b) Relative Citation Ratio for U.S. Medical Student Authored Pubications by Year**

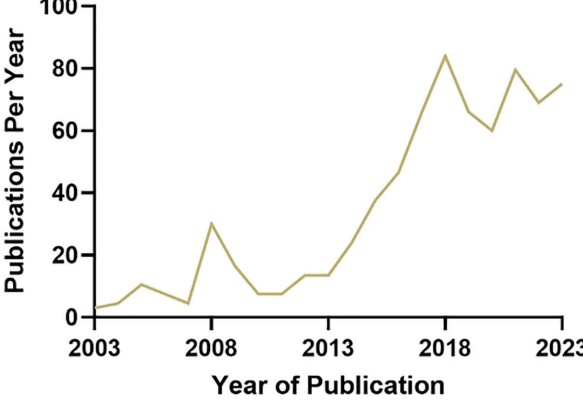

**c) Number of International Medical Student Authored Publications by Year**

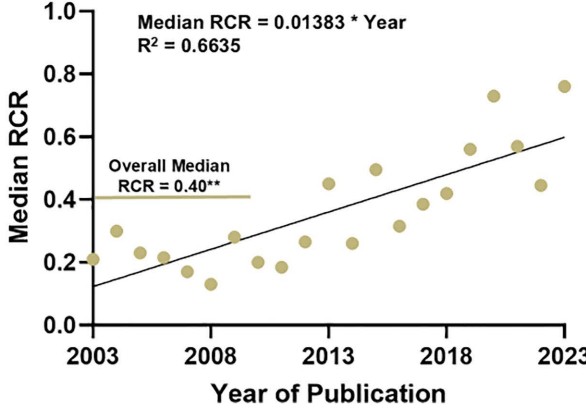

**d) Relative Citation Ratio for International Medical Student Authored Pubications by Year**

**Fig 7. Trends in medical student authored research in U.S. and International Cohorts (2003-2023).** (a) Research output from U.S. medical student authored publications (2003–2023). (b) Research impact in Relative Citation Ratio (RCR) for U.S. medical student authored publications. (c,d) Corresponding data from international medical student authored publications over the same period. Note the increase in publication volume and RCR for both groups. While international medical students accounted for more publications overall, U.S. medical student publications generated a significantly higher median RCR.

differences may be attributable to research environment and visibility. Among international contributors, the United Kingdom emerged as the highest volume contributor by 2023, followed by Canada, India, and Brazil (Fig 8). In terms of article impact, Canada had the highest median RCR (0.59) followed by Iran (0.49), Australia (0.40), and Brazil (0.35). All major contributing countries demonstrated upward trends in publication volume over the study period, indicating a global increase in medical student research activity.

## Discussion

Our study demonstrates that medical student research has undergone dramatic growth in both volume and quality between 2003 and 2023. While residents produced nearly four times more publications overall (5,365 vs. 1,443 articles),

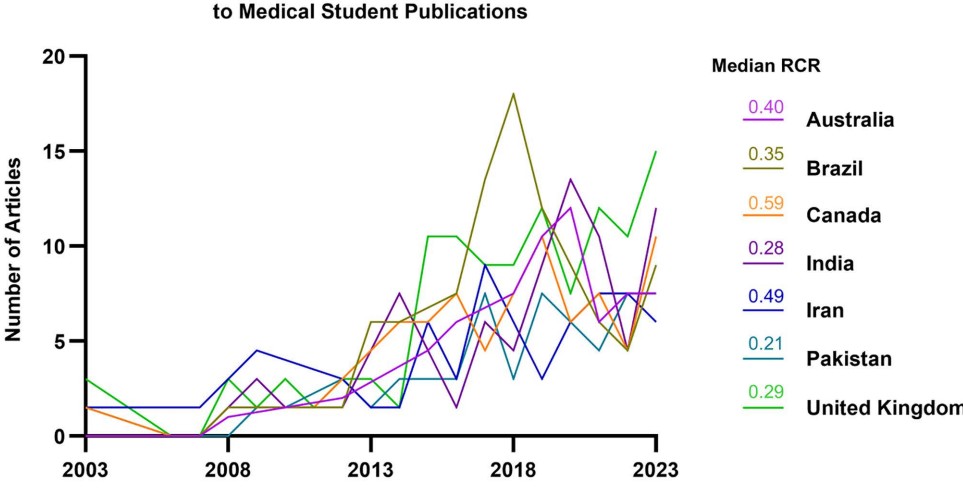

**Fig 8. Global trends in medical student research authorship (2003–2023).** Number of medical student–authored publications from the top eight international contributor countries in our cohort (2003–2023). All countries demonstrated an upward trend in publication volume over time, with the United Kingdom, India, Brazil, and Canada being the highest contributors in 2023.

medical student research demonstrated comparable citation performance, with a median RCR of (0.47 vs. 0.28, p = 0.02). By 2023, the RCR of medical student-authored publications (0.8) approached the benchmark for a median NIH-funded article (RCR = 1.0). Furthermore, case reports, although a common and more accessible form of research, have significantly lower citation potential [17]. Exclusion of case reports in our analysis demonstrates that both medical student (0.98) and resident publications (0.87) approximate a RCR of 1.0. Taken together, these findings show that medical student-authored research over the past 20 years has achieved an impact comparable to resident-authored work and has contributed to a growing body of influential publications worldwide.

A 2013 study from Sri Lanka published in *BMC Medical Education* identified a total volume of 350 PubMed and Scopus-indexed medical student-authored articles published globally between 1980 and 2010, averaging approximately 11 articles per year and reporting an exponential increase in publication volume over time [15]. However, most of those articles (59.1%) had never been cited, and the mean number of citations per article was only 4.5. The most common research areas were Psychiatry, General Medicine, Medical Education, Oncology, and Community Medicine, with review articles (13.7%), cross-sectional studies (13.4%), case reports (12.3%), case-control studies (11.7%), and cohort studies (10.6%) being the most frequently published article types. Our study observed a dramatically different volume, with 1,443 medical student publications from 2003 to 2023, averaging approximately 69 articles per year: over six times the annual output reported in the 2013 study. Our dataset also demonstrated markedly superior citation performance: only 10% of articles remained uncited, and the average number of citations per publication was 16.04 (Fig 3). These substantial differences likely reflect both the different time periods studied and the growing institutional support for medical student research, particularly in the U.S. where research experience has become increasingly valued for residency and faculty appointments [2].

A 2019 Michigan study examining medical student and resident-authored publications between 2002 and 2016 in *Academic Medicine* analyzed 4,635 articles, concluding that trainee-authored publications significantly increased over the duration of the study [18]. The study reported that medical student authorship increased from 4.4% to 10.3% of all articles published in *Academic Medicine*, while resident authorship increased from 5.2% to 9.8% [18]. Our analysis shows a

similar, although more dramatic (more than 10-fold) increase in medical student and resident-authored publications across this period. Moreover, we analyzed the scholarly impact of trainee publications using citation metrics, while this 2019 study primarily examined authorship trends. Additionally, unlike our analysis, this study only included publications from a single journal, while we queried all PubMed-indexed articles meeting our inclusion criteria. This previously published approach demonstrates the power of granular indexing for specific journals or topics of interest, but it may be less representative of broader trends in medical trainee research due to gaps in coverage across geography, subject matter, and study design. Our more heterogeneous sample may provide greater clarity on these dimensions of medical student and resident research.

A 2023 study in Virginia also utilized affiliation-based searching in PubMed indexed articles from 2008 to 2022 to examine trends in medical student publication volume and NRMP specialty-specific indexes of productivity [9]. They found that medical student publications have linearly increased over time, reporting over 300 publications per year in 2022 compared to 25 per year in 2008, which followed a corresponding increase in the NRMP's average research items per matched applicant, from 4 in 2008–11 in 2022 [19]. This represents a nearly 12-fold increase in volume, which corroborates well with the 15-fold increase in volume observed in our analysis. However, they did not account for the dimension of quality, rather hypothesizing that increasing research volume came at the expense of research impact. In their article, Elliot et al. again cited results from Wickramasinghe et al., which found that 59% of medical student publications were never cited [9,15]. Notably, Wickramasinghe et al. analyzed articles from 1980–2010, meaning only three years coincided with the study period in the study conducted by Elliot et al., [9,15] highlighting the need for updated data on the impact of student research over the last 20 years. We address this gap by examining quality metrics that have demonstrated that medical student research has high citation impact globally. Building on these findings, we also provide specialty- and country-specific RCR benchmarks within the *ImpactLens* tool as a means for both medical students and selection committees to evaluate research quality on an article-by-article basis, accounting for differences in specialty and regional publication trends (S2 File). By offering a field-normalized, objective measure of scholarly impact, these benchmarks can help identify high-quality trainee research and facilitate more equitable and transparent evaluation of scholarly productivity.

Our study is not without limitations. First, we limited our search to English-language articles indexed in PubMed, which may limit the generalizability of our findings to non-English publications or those not captured in PubMed. Yet, despite these constraints, our approach captured over 750 international medical student-authored articles from over 20 countries, providing data that are broadly representative of global research activity. Second, while we utilized affiliation-based queries to identify medical student- and resident-authors, this approach is not comprehensive and is a limitation seen in existing literature [20]. Misclassification is possible due to inconsistencies in how affiliations are reported, and not all authors include a clear "medical student" designation – likely resulting in missed publications. Although some selection bias is therefore unavoidable, utilizing a systematic approach to querying yielded a randomized and representative dataset within the constraints of available indexing methods, including a diverse sample of publication types from both U.S. and international institutions. This affiliation-based indexing method, which demonstrated 96% specificity for medical student articles and 82% specificity for resident articles, provides a reproducible framework for identifying trainee-authored publications (S1 Fig). To improve future bibliometric analyses of medical student research, we recommend authors include affiliation terms such as "medical student," "MD candidate," or "medical resident" rather than relying solely on individual school titles. Third, we did not analyze changes in medical student research volume and impact trends after USMLE Step 1 transitioned to pass/fail in 2022. Our study may be undermeasuring the influence of this change given the brief time between 2022 and the end of our article inclusion (2023). Lastly, our analysis did not directly quantify the influence of medical student or resident authors on their respective publications by authorship position or a corresponding metric, nor did we take into account the role of the primary investigator, although we did find that medical students were listed as 1st authors in (635/1445) or 44% of their publications. Additionally, per ICMJE policy, which most included journals follow, all persons listed as authors must have made substantial contributions to the project and development of the manuscript, meaning

regardless of author position, students and/or residents listed must have performed meaningful work [21]. Future studies may aim to delineate the authorship positions and associated impacts on RCR based on primary contributor.

## Conclusion

In recent decades, medical student research output has grown substantially. To our knowledge, this is among the first studies to systematically evaluate medical student research volume and impact using the Relative Citation Ratio and to provide specialty- and country-specific citation benchmarks for medical student publications. These results underscore the influence of student-led scholarship and support continued investment in mentorship, infrastructure, and institutional support. Future studies should seek to examine the trends in volume and scholarly impact of medical student research before and after the 2022 transition of USMLE Step 1 to pass/fail, identify the educational and policy interventions that have contributed to improved trainee research, and explore whether differing approaches to evaluating trainee research like *ImpactLens* lead to better identification of students with the greatest potential for impactful scholarship. Undoubtedly, an understanding of these factors will only grow in importance as medical education continues to embrace a research culture. Nonetheless, our bibliometric analysis demonstrates that medical students contribute impactful research that is frequently cited and built upon. As academic medicine continues to evolve, the scientific voices of future physicians are already making their mark.

## Supporting information

**S1 Fig. PubMed search queries.** PubMed Search queries for medical student and resident-authored articles. The number of articles identified at each step, filters used, and accuracy of each search strategy is identified to enhance reproducibility.
(TIF)

**S2 File. ImpactLens.** ImpactLens categorizes all 1,443 included medical student-authored articles over the last 20 years to help medical student and faculty researchers benchmark medical student-authored articles against others in their field and country of origin. To share new articles, visit our web-app at: https://impactlensgit.netlify.app/.
(XLSX)

**S3 Appendix. List of medical student articles.** Provided is our list of medical student-authored articles indexed from PubMed and analyzed through the NIH iCite tool to generate the data presented in this manuscript.
(XLSX)

**S4 Appendix. List of medical resident articles.** Provided is our list of medical resident-authored articles indexed from PubMed and analyzed through the NIH iCite tool to generate the data presented in this manuscript.
(XLSX)

## Acknowledgments

The authors report no disclosures, funding, or other conflicts of interest. Ethical approval was not applicable. We would like to acknowledge three members of our team: Richard Tuttle, Sydney Huston, and Diana Kernen for their contributions in data collection for this manuscript.

## Author contributions

**Conceptualization:** Christian J. Hausner, Michelle Yi, Meet S. Patel.

**Data curation:** Christian J. Hausner, Michelle Yi, Meet S. Patel.

**Formal analysis:** Christian J. Hausner.

**Investigation:** Christian J. Hausner, Michelle Yi, Meet S. Patel, Jack T. Franchino.

**Methodology:** Christian J. Hausner, Michelle Yi, Meet S. Patel, Charles S. Day.

**Project administration:** Christian J. Hausner, Michelle Yi, Meet S. Patel, Jack T. Franchino, Charles Day.

**Resources:** Charles S. Day.

**Software:** Christian J. Hausner.

**Supervision:** Christian J. Hausner, Charles S. Day.

**Validation:** Christian J. Hausner, Jack T. Franchino.

**Visualization:** Christian J. Hausner.

**Writing – original draft:** Christian J. Hausner, Michelle Yi, Meet S. Patel, Jack T. Franchino.

**Writing – review & editing:** Christian J. Hausner, Jack T. Franchino, Charles S. Day.

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
