## [Decision Letter · Decision Letter 0]

7 Jan 2026

Dear Dr. Day,

Thank you for submitting your manuscript to PLOS ONE. After careful consideration, we feel that it has merit but does not fully meet PLOS ONE’s publication criteria as it currently stands. Therefore, we invite you to submit a revised version of the manuscript that addresses the points raised during the review process.

We look forward to receiving your revised manuscript.

Kind regards,

Inge Roggen, M.D., Ph.D.

Academic Editor

PLOS One

Journal Requirements:

2. Please include a copy of Table 4 which you refer to in your text on page 14.

Reviewers' comments:

Reviewer's Responses to Questions

**Comments to the Author**

1. Is the manuscript technically sound, and do the data support the conclusions?

Reviewer #1: Yes

Reviewer #2: Yes

2. Has the statistical analysis been performed appropriately and rigorously?

Reviewer #1: Yes

Reviewer #2: Yes

3. Have the authors made all data underlying the findings in their manuscript fully available?

Reviewer #1: Yes

Reviewer #2: Yes

4. Is the manuscript presented in an intelligible fashion and written in standard English?

Reviewer #1: Yes

Reviewer #2: Yes

Reviewer #1: First I would like to congratulate the authors for doing a comprehensive study. This study includes 1,443 medical student-authored and 5,365 resident-authored articles—a substantial dataset providing statistical power. Further, the 20-year analysis (2003–2023) captures meaningful trends, allowing examination of how medical student research productivity has evolved in response to changing educational emphases and residency application competitiveness.

1. However, while the study includes international data, the dominant cohort and highest-impact publications are from the United States. The findings may not generalize well to medical education systems in other countries with different training structures, research culture, or publication incentives.

2. The authors acknowledge that using affiliation-based PubMed queries to identify medical students and residents is "not comprehensive" and can lead to misclassification

3. The analysis is limited to English-language articles indexed in PubMed only, which excludes non-English publications and research from databases not captured in PubMed. This significantly limits generalizability to global medical student research; so try to expand database coverage beyond Pubmed

4. The study does not quantify the influence of medical student or resident authors by their authorship position (first author, middle author, corresponding author) or account for the role of the primary investigator. This means we cannot determine whether students were the driving force or minor contributors to the work. Contact corresponding authors or institutions directly to confirm authorship status and role (first author, senior author, mentee vs. mentor-led). This would reduce affiliation-based misclassification and clarify the student's actual contribution level.

5. Present separate analyses for first-author vs. other-author positions. Students as first authors likely drive research conception and execution more substantially than when appearing as co-authors, making this distinction important for interpreting the findings.

Reviewer #2: This paper systematically compares the research output of medical students and resident physicians based on 20 years of large-scale bibliometric data. The topic is relevant to current research, the data is substantial, and the methodology is generally transparent and reproducible. However, the current manuscript has several key issues requiring significant revision to enhance scientific rigor.

1. The paper frequently interprets citation counts and RCR (Research Comparison Rate) directly as evidence of research "quality" or "high-level contribution." However, RCR essentially reflects relative academic influence, not methodological rigor, originality, or research quality itself. Authors need to clearly distinguish between "academic influence" and "research quality" in the abstract, discussion, and conclusion.

2. Several core conclusions rely on the analysis results after excluding case reports. However, given the high proportion of case reports in the research of trainees, this exclusion may introduce selection bias and unintentionally inflate the overall impact index.

3. While this paper focuses on the research contributions of medical students, the analysis does not address the impact of authorship order on the paper's contribution. Authorship alone cannot determine an author's actual academic role in the research.

4. Although RCR is a field-normalized indicator, systematic differences still exist between different research types (e.g., reviews, meta-analysis, and original research). The current manuscript has not fully discussed the potential impact of differences in research type composition on RCR results.

5. Using only PubMed data may introduce subject, language, and regional biases, and the impact on international comparison results requires further discussion.

6. Some differences (the RCR of papers by US medical students is higher than that of international medical students (0.58>0.40, p = 0.02)) are statistically significant, but not in terms of research ability itself. Their actual educational or policy implications have not been fully explained.

7. The figures and tables in this paper are rich in information but not clear enough, and their correspondence with the core arguments of the text is not direct enough.

**Do you want your identity to be public for this peer review?** For information about this choice, including consent withdrawal, please see our Privacy Policy

Reviewer #1: **Yes:** Anamika Gulati

Reviewer #2: **Yes:** Shuai Liu

---

## [Author Response · Author response to Decision Letter 1]

26 Jan 2026

We would like to thank the editor and reviewers for their comments and thoughtful assessment of the manuscript. All responses to editor and reviewer comments can be found in the supplemental document on page 46. A manuscript with track changes can be found included beginning on page 47 of the pdf. Thank you all.

---

## [Editor Report · Decision Letter 1]

2 Feb 2026

Medical Student Research Productivity and Scholarly Impact: A 20-Year Bibliometric Comparison with Medical Residents

PONE-D-25-59289R1

Dear Dr. Day,

We’re pleased to inform you that your manuscript has been judged scientifically suitable for publication and will be formally accepted for publication once it meets all outstanding technical requirements.

Kind regards,

Inge Roggen, M.D., Ph.D.

Academic Editor

PLOS One
---

## [Editor Report · Acceptance letter]

PONE-D-25-59289R1

PLOS One

Dear Dr. Day,

I'm pleased to inform you that your manuscript has been deemed suitable for publication in PLOS One. Congratulations! Your manuscript is now being handed over to our production team.

Kind regards,

on behalf of

Prof. Inge Roggen

Academic Editor

PLOS One